# Initiatives to Reduce the Content of Sodium in Food Products and Meals and Improve the Population’s Health

**DOI:** 10.3390/nu15102393

**Published:** 2023-05-19

**Authors:** Karolina Jachimowicz-Rogowska, Anna Winiarska-Mieczan

**Affiliations:** Department of Bromatology and Food Physiology, Institute of Animal Nutrition and Bromatology, University of Life Sciences in Lublin, Akademicka 13 St., 20-950 Lublin, Poland; anna.mieczan@up.lublin.pl

**Keywords:** salt, salt intake reduction, sodium, cereal products, food education, non-communicable diseases, hypertension, sodium sensitivity, food manufacturing, salt substitutes

## Abstract

Table salt is the main source of sodium (Na) in the human diet. Excessive supply of Na in a diet is strongly linked to many non-communicable human diseases, such as hypertension, obesity and stomach cancer. The World Health Organization recommends that daily intake of salt in adult diets should be kept below 5 g/person/day, which corresponds to 2 g Na/person/day. However, on average, adults consume about 9–10 g/person/day, and children and young people about 7–8 g/person/day. Initiatives to reduce salt intake include modifications of food composition in collaboration with the food industry, education of consumers, salt marking on foodstuff labels and taxation of salt. A need also exists to educate society so that they choose low-sodium products. In view of the food technology and amount of salt intake, the most important and the easiest change to make is to reduce the content of salt in baked goods. This paper analyses the results of surveys regarding strategies to reduce salt content in food products and considers multifaceted initiatives to reduce salt intake as a possible efficient method of improving the population’s health status.

## 1. Introduction

Recently, we have observed a growing consumer interest in food quality and safety as well as the effect it has on human health. Consumers wish food was healthier and contained fewer ingredients believed to have adverse impact on health [1]. Sodium chloride plays a significant role in food processing, having several functions such as: sensory (shaping taste), texture-forming (affecting water/fat binding capacity) and bacteriostatic (inhibiting microbial growth) functions [2].

According to Liem [3], salty taste preference develops due to multiple exposures to salty foods. Infants develop salty taste gradually. Most likely, below three months of age, they are incapable of tasting salt. However, when infants start recognising saltiness, this becomes their preferred taste. Similar to their preference for sweet taste, children prefer salt at higher concentrations than adults do [3].

Salt (table salt, sodium chloride, NaCl) consists of sodium (Na^+^) and chloride (Cl^−^) ions. Approximately 90% of sodium and chlorides in the diet are derived from salt [4]. Since salt is commonly added to various products and dishes, sodium deficiency in the diet is very rare. By contrast, excessive sodium intake is a much more frequent problem, which increases the excretion of calcium with urine and aggravates the risk of hypertension [5]. When 1 g of Na is consumed, as much as 26 mg of Ca will be excreted with urine, which is particularly important for the commonly recorded Ca deficiency in the diets of Polish people [6]. Long-term commitment to reducing Na intake is a challenge to the general population. According to Global Burden of Disease data [7], 1.89 million deaths from cardiovascular diseases (CVD) per year worldwide are associated with excessive intake of Na. A meta-analysis of studies conducted from 1946 to 2020 showed that Na reduction following applicable recommendations contributed to reduction in blood pressure by about 0.4 mmHg in normotensives and by about 4 mmHg in hypertensives [8]. The World Health Organization (WHO) recommends limiting salt intake to 5 g (2 g of sodium) per day for adults [9], whereas for food marking purposes, the reference intake of salt was determined as 6 g day [10]. The content of salt and sodium in products or diets should be calculated based on the assumption that 1 g of sodium corresponds to about 2.5 g of salt [11]. The European Commission has published a nutrition and health claims register that contains the list of claims that can be made about food. There is one permitted health claim for food with low or reduced sodium content: “Reducing consumption of sodium contributes to the maintenance of normal blood pressure” [12]. Sodium restriction recommendations concern both patients with hypertension and those who want to lead a healthy lifestyle [13].

Systematic reviews and meta-analyses imply considerable epidemiological evidence that consuming wholegrain products which supply at least dietary fibre is associated with a lower risk of developing non-communicable diseases related to the diet [14]. By contrast, cereal products supply considerable amounts of sodium, and table salt is used to prepare them. For example, added salt is technologically significant to the bread baking process. Dough with an excessively reduced salt content may tend to over ferment, and yeasts can produce too much carbon dioxide, which deteriorates the texture of baked goods [15]. Acceptance of low-salt baked goods by consumers is also essential. Consumers will not even note a slight reduction in the amount of salt in a product’s formula (ca. 0.1–0.2%). The human body gradually adapts to and fully accepts less salty foods, so the amount of salt added to baked goods should be gradually reduced [16].

Many initiatives are undertaken around the world to reduce the consumption of salt [17]. These measures are a major challenge for food producers as they require time and financial expenditure to modify technological processes. While such initiatives generate costs for food manufacturing and require public money for health campaigns, they reduce the availability of unhealthy foods and may result in lower healthcare system costs for chronic diet-related diseases [17]. According to Zhang et al. [18], in the United States, reducing individual sodium intake to 2300 mg/day from the current level could potentially save $1990.9/person per year for hypertension treatment. Baked goods constitute an important source of sodium since most people eat them every day in large amounts [19,20,21]. This points to the need to reduce salt content in the baking industry for reasons of consumer health [22]. Our study showed that baked goods cover 48.2% of the Na requirement of an adult person [23]. Generally, bread contains relatively low levels of added salt (on average 2% of all the ingredients), but people consume considerable amounts of baked goods, which—as a result—makes up between 35% and 50% of sodium intake [15,22].

This paper analyses the results of surveys regarding strategies to reduce salt content in cereal products and considers multifaceted initiatives to reduce salt intake as a possible efficient method of improving the population’s health status. The literature review was conducted in April 2023 using the PubMed, Scopus, Web of Science and Google Scholar literature databases. The databases were searched for both joint and separate instances of the keywords, e.g., “salt”, “salt intake reduction”, “sodium”, “sodium:potassium ratio”, “cereal products”, “food education”, “non-communicable diseases”, “hypertension”, “sodium sensitivity”, “food manufacturing”, and “salt substitutes” in both the English and Polish languages (Figure 1). The search was narrowed to papers published within the last 10 years. Texts were thoroughly analysed with a view to selecting the most pertinent publications. Based on the titles and synopses, articles unrelated to the substantive criteria were excluded, and the remaining original and review papers were intensely analysed to identify the most pertinent publications. Ultimately, a total of 343 publications were reviewed, of which 166 were used: 72 research reports and 94 reviews and meta-analyses (Figure 1). In this monograph, ‘salt’ refers to sodium chloride only.

## 2. Perception of Salty Taste, Salt Craving, and Sodium Sensitivity

Taste bud cells, intracellular signals and signal pathways responsible for perceiving salty taste have still not been fully explored [24,25]; however, how the sense of taste detects salt has been intensively investigated. Detection of salt by the sense of taste is of fundamental importance for the intake of salt and tissue homeostasis [26]. Nomura et al. [27] identified taste cells responsible for Na^+^-selective, attractive components of salty taste and for intracellular signal transduction. The above-quoted authors reported that the amiloride-sensitive epithelial sodium channel (ENaC) used as a salt receptor is co-expressed with the voltage-gated channels releasing adenosino-5′-triphosphate (ATP)-calcium homeostasis modulator 1/3 (CALHM1/3) in a subset of taste cells, and these cells mediate in amiloride-sensitive transduction of salty taste. It was demonstrated that components sensitive and insensitive to amiloride differ in terms of being selective to salt (the amiloride-sensitive component is NaCl-selective in relation to potassium chloride (KCl) and that they are mediated by various cells within a taste bud (the amiloride-sensitive component is restricted to the front part of the tongue) [26]. In order to control the correct intake of sodium, many mammals are equipped with two salt tasting mechanisms: sodium taste and high-salt taste [25].

The ions of Na^+^ in the mouth find their way into taste receptor cells through ENaC channels (without changing the intracellular concentration of Ca^2+^), leading to suprathreshold membrane depolarisation, generating action potentials, creating a channel synapse with afferent neurons, which involves a voltage-gated channel releasing neurotransmitters that comprise CALHM1 and CALHM3—CALHM1/3. The CALHM1/3 channels open in response to strong membrane depolarisation and release ATP to afferent nerve fibres (Figure 2). Thus, CALHM1/3 channels mediate in puringenic neurotransmission of the sodium taste. Salty cells can be unambiguously identified owing to the co-expression of CALHM1/3 and ENaC only, which is required for amiloride-sensitive transduction of salty taste. All sodium taste signalling stages depend on voltage and not on Ca^2+^ signals. The elimination of ENaC in CALHM1-expressing cells, as well as global CALHM3 deletion, abolishes amiloride-sensitive neural responses and attenuates behavioural attraction to NaCl [27].

Perception disorders accompanying various medical conditions are deemed a disorder of taste receptors and can be caused by an infection spreading to cranial nerves responsible for transmitting taste; this was found, for instance, in COVID-19 patients who reported hypersensitivity to salty taste [28]. Mice deprived of the gene responsible for synthesising Engrailed-2 (En2—transcription factor critical to the development of neurons) showed a sensory disorder with a measurable influence on the perception of taste and, in particular, hypersensitivity to salty taste [29]. Salt craving, also referred to as sodium appetite, is mostly regulated by the hypothalamus and periventricular neurons [30].

In the course of individual development, the foliate papillae in the mucous membrane are reduced, which leads to a partial loss of taste sensitivity. It was demonstrated that, as we age, the number of taste buds is reduced even to about 30% of the starting value [31]. In parallel epithelial keratinisation progresses, which significantly affects the function of the sense of taste. Such dysfunction can also result from damage to the taste pathway in the nervous system. It was demonstrated that irregularities in the areas of the brain critical to processing olfactory information could be the main reason for dysosmia in the elderly population [32]. It is estimated that smell disorders occur in 50% of COVID-19 patients [33].

Persons experiencing blood pressure fluctuations under the influence of sodium in the diet are often referred to as ‘sodium-sensitive’. Sensitivity to sodium is more frequent in people who have hypertension (30–50% of cases compared to 15–25% cases without hypertension), obese, suffering from diabetes, elderly, prematurely born, SGA babies (Small for Gestational Age), Afro-Americans, people with the metabolic syndrome and women [34,35,36]. The wide-ranging international study INTERSALT, conducted at 52 centres in 32 countries, corroborated the correlation between the amount of excreted sodium (being a measure of daily intake of this element) and the blood pressure value [37]. However, at the same time, genetic conditions can have an impact on individual sensitivity to sodium [38], so an excess of sodium in the diet does not increase the risk of developing hypertension to the same extent for everyone, which could explain why different populations have different sensitivity to salt [39]. The kidneys are one of the main organs destroyed by hypertension. In contrast, it is commonly believed that renal disorder plays a central role in the pathogenesis of hypertension in sodium-sensitive patients [35]. Furthermore, low weight at birth (<1000 g) and preterm birth increase the risk of hypertension for adults [40]. Interestingly, those who are sensitive to sodium show a stronger reaction to reducing salt in the diet than those resistant to sodium [4]. The diets of most sodium-resistant people can be rich in salt with no risk of developing salt-induced hypertension [41].

A complete and efficient treatment-supporting diet now recommended to overweight, hypertensive, hypercholesterolemic and hyperlipidaemic patients is DASH (Dietary Approaches to Stop Hypertension). Its main assumptions include at least four or five meals a day composed of unprocessed fruits and vegetables, cereal products, low-fat dairies, fish and nuts. The total supply of salt with the DASH diet should not exceed 2300 mg sodium/day [42]. For DASH users, salt intake increased up to 8.8 g/day did not significantly increase their blood pressure levels [41].

## 3. Actual versus Recommended Intake of Sodium in Various Populations

People generally know they consume too much salt but are not aware that they exceed the recommended intake more than two times [43]. Currently, in European countries, the average intake of salt by women is 7.3–10 g/day, and by men, 9.4–13.3 g/day [44], which is 1.5–2.7 times higher than the recommended daily intake. This may be due to the lack of awareness about sodium being present in food products consumed even more than once a day, such as bread and other cereal products [45]. Such a high intake of salt can contribute to the increased number of hypertensive patients in Poland [13].

According to the WHO Sodium Intake Reduction Report [17] and involving data from Global Burden Disease report [7], on average, people consume 4310 mg sodium per day (10.78 g of salt per day), which is double the recommended amount. The WHO proposes intake <2000 mg of sodium (equivalent to <5 g of salt) per day in adults, which is required to reduce the risk of hypertension and cardiovascular diseases. The WHO regional sodium intake in 2019 estimates range from the lowest 2687 mg/day (6.2 g/day salt) in the African Region to the highest 6247 mg/day (15.6 g/day salt) in the Western Pacific Region. The estimated intake for China is 6954 mg/day sodium, which is likely to be influencing the Western Pacific Region mean.

Actual sodium intake by the WHO Member States all over the world is presented in Sodium Intake Reduction Report [17]. Average American and Canadian diets consist of, respectively, 3400 mg and 2950 mg of Na per day, far exceeding the dietary recommendations of the WHO. The average sodium content in a daily food ration in Poland was higher than in the USA and amounted to 4357 mg/day, which considerably exceeds Polish sodium intake standards [46,47]. It is estimated that Polish people consume about 11.1 g of salt per day. The situation in Germany looks better but still exceeds standards; German people consume about 3410 mg sodium (8.7 g salt) per day. The average salt intake in Ireland was 7.3 g/day. Malaysians consume approximately 4134 mg of sodium a day, while most sodium in the diet derives from ready-to-eat food and sauces [48]. However, above mentioned intakes are still lower than reported for other countries with varied diets such as Romania (5075 mg of sodium per day), Czechia (5112 mg of sodium per day), Hungary (5646 mg of sodium per day), and China (6954 mg of sodium per day). All the Member States exceed the recommendations of the WHO. Selected sodium intakes (the lowest and the highest) and salt equivalents in various countries of the world are shown in Table 1.

Childhood and adolescence are two key periods in human life during which eating habits and preferences develop [49,50]. A limited intake of salt in childhood is likely to prevent the development of hypertension and its related consequences in adult life [51]. The intake of sodium by the infant was evaluated, for instance, in the studies by Girardet et al. [52]. Their results imply that most infants consume an excessive amount of sodium at an early stage of their lives. According to these studies, the sodium intake of infants is often higher than the values recommended by national and international guidelines, which results in about 4–8 times higher intake of sodium [53]. In the USA, using a 24-h diet reminder used in the NHANES 2011–2012 survey with 2142 children and adolescents (6–18 years old), salt consumption was 8.1 g per day [54]. In Spain, a study carried out in 2014 with 205 children between 7 and 11 years old using 24-h urine sodium excretion, reported 7.8 g of daily intake of salt [55]. In Italy, a study covered 1424 children and adolescents between 6 and 18 years old from whom 24-h urine samples were collected, finding a salt intake of 7.1 g per day [56]. Other studies carried out in Portugal [57], using dietary registration and 24-h urinary excretion, reported 2658 mg/day sodium excretion. In total, 62% of Swiss children had salt excretions above the maximum intake recommendations (≥2 g for up to 6 years and ≥5 g per day from 7 to 16 years) [58]. On average, children and adolescents consumed from 1.6 to 1.8 mg sodium/kcal, a much higher amount than 1 mg sodium/kcal, proposed by Guenther et al. [59].

## 4. Impact of Excessive and Insufficient Intake of Sodium on the Human Health

Almost two million deaths each year are associated with excessive sodium intake [7], a well-established cause of raised blood pressure and increased risk of CVD [60,61]. Reducing sodium intake is one of the most cost-effective ways to improve health and reduce the burden of non-communicable diseases, e.g., hypertension (systolic blood pressure ≥ 140 mmHg and/or diastolic blood pressure ≥ 90 mmHg), coronary artery disease, cardiac infarction, stomach cancer, obesity, osteoporosis, increased extracellular fluid volume, oedemas, chronic renal failure and higher general mortality rate [11], as it can avert a huge number of deaths with very low costs. The WHO recommends several sodium-related best buy interventions and other measures as practical actions that should be undertaken immediately to prevent CVD and its associated costs. Globally, an estimated 2.2 million and 7 million CVD deaths could be averted by 2025 and 2030, respectively, if countries implemented the policies suggested by WHO actions. This equates to a 3.1% reduction in CVD death, globally, by 2030 [17].

Studies confirmed that a high sodium intake is the main cause of CVD. Outcomes obtained by Emmerik et al. [62] show that a high sodium intake during the first six months after birth can also trigger negative effects for health, such as raised blood pressure levels. Infants exposed to large amounts of sodium tend to prefer the salty taste and are susceptible to renal disorders in adult life. In addition, mean sodium intake was estimated for more than 2000 respondents aged between 6 and 18 years in the National Health and Nutrition Examination Survey. Schoolchildren in the USA consume 3279 mg of sodium a day, which is above the recommended level [63]. Cotter et al. [64] obtained similar results for children aged 10–12 and found that more than 90% of the students consumed more salt than was recommended. Preparing infant formula using mineral water low in sodium or provide breast milk is recommended, along with reducing salt intake with foodstuffs and avoiding adding salt in meal preparation [52].

The latest evidence points to a considerable shift In the consumption pattern of high-sodium foods during the COVID-19 pandemic in the populations of different countries [65], including processed and canned food products that are bought ahead and quick meals. The study of the French population (*n* = 11,391) shows that, during the first wave of the pandemic, consumption of salted food surged by 28.4% [66]. Excessive sodium intake with the diet contributes to the development of chronic diseases, such as hypertension, cardiovascular diseases and kidney diseases, thus entailing a risk of occurrence or exacerbation of comorbidities after contracting COVID-19 [65]. Society should be aware that maintaining a healthy diet during quarantine or isolation is as important as infection control and preventive measures to mitigate health risks in a population [67].

Knowledge, beliefs and perceptions about salt and sodium and their role in health and illness can differ depending on the community. D’Elia [68] observed that 86% of patients were aware that excessive salt intake adversely affects human health, but only 44% of them believed it was worth reducing salt consumption. Sodium intake reduction is an important strategy for reducing high blood pressure and preventing cardiovascular diseases, both for adults [8] and children [8].

However, it was feared that reduced sodium intake would adversely affect insulin resistance, the lipid profile of blood, the level of catecholamines and cardiovascular disease risk factors and the general health status due to excessive lowering of the level of this element in the blood. Hyponatremia occurs when the concentration of sodium in blood serum is lower than 135 mmol/L. The aetiology of hyponatremia is multifaceted, and consuming poor quality foods that contain insufficient amounts of salt can be one of its causes [69]. A counterpoint to the current recommendation for low Na intake is included in the article by O’Donnell et al. [70], which suggests that a specific low Na intake target (920 mg/day) for individuals may be infeasible and have uncertain effects on other dietary factors as well as unproven effectiveness in reducing CVD. A moderate range of dietary Na (920–1840 mg/day) is not associated with increased CVD risk, but the risk of CVD increases when Na intakes exceed 2000 mg/day. It is worth noting, however, that further Na reductions to 1500 mg/day or less are not advised among high-risk groups (i.e., individuals with heart failure, diabetes, kidney disease and CVD), as such low intakes have been associated with adverse CVD outcomes [71]. It is important that high risk groups have a properly balanced diet because both too much sodium (excessive use of table salt and eating salt-rich processed foods) and extremely too little (using low sodium salt substitutes) may not be effective in reducing CVD risk [72,73]. However, the latest reviews of studies involving adults show that there is insufficient evidence to imply that reducing sodium in the diet has a negative impact on blood glucose level measurements, insulin resistance, the lipid profile of blood or catecholamine levels in interventions lasting at least four weeks [71,74].

## 5. Sources of Salt in the Human Diet

Food products with a considerable sodium chloride content include table salt and food seasoning, baked goods, cold meats and cheeses. In developed countries, excessive intake of sodium is mostly (75–80%) due to consumption of processed food and ready meals, 5–10% occurs naturally in food products that form part of the diet and the remaining 10–15% derives from salt added in cooking or on the plate [75,76]. An exception to that rule is the inhabitants of Czechia, Poland and Romania (European developed countries), for whom salt added in cooking is the most important source of sodium in the diet [77]. In many European countries, due to a lack of time, homemade meals are rare, so only a small percentage of sodium intake derives from salt added in cooking [4]. More than one-third (34%) of Swiss children reported adding salt to their meals on the plate [58], but this figure was even higher for Costa Ricans (49.4%) [78]. In the quoted study, nearly 39% of the participants said they usually add at least one pinch of salt to their meals one or two times a week; in turn, one in ten children from Costa Rica declared that they added a pinch of salt even three or four days in a week. However, reducing sodium intake by avoiding adding salt on the plate and table salt in cooking may not be particularly efficient since most sodium in the diet derives from processed foods and restaurant meals [11]. In many high-income countries, and increasingly in low- and middle-income countries, a significant proportion of sodium intake can be attributed to take-away and out-of-home foods. One of the WHO goals is to reduce sodium intake through reducing sodium in meals or snacks consumed outside of the home and even restricting the availability of saltshakers in service areas. Therefore, it is important to implement public food procurement and service policies to reduce sodium content in food served and sold. Such policies are most common in the WHO regions of the Americas, Europe and Western Pacific, and more frequently implemented in higher income groups [17]. Processed foods, such as fried chicken, pizza, tomato sauces, fried ready baked goods, sandwiches, burgers and sausages were identified as entailing the risk of diet-related diseases (such as obesity, diabetes and hypertension) due to their high content of fat and/or sodium [79]. High salt content is typical of fast foods. A typical fast food lunch consisting of a sandwich and chips and dip (usually ketchup) contains 4.5 g of salt, which corresponds to as much as 90% of the recommended intake. Another popular fast food, doner kebab, contains from 4.0 g to 8.4 g of salt, and pizza containsfrom 7.0 g to 12.8 g. So-called instant soups should also be mentioned as they contain up to 4.1 g of salt per serving [11]. Nevertheless, international restaurants still target such meals at children, adolescents, and whole families [80].

The range of sodium content in food product categories is often extreme, both very high and permissibly low. This follows clearly from the comparison of salt content in fresh products and their processed equivalents such as fresh green (string) beans vs. canned beans, raw cabbage vs. sauerkraut (fermented cabbage), tomatoes vs. ketchup, raw ham vs. prosciutto, fresh herring vs. pickled herring, and low-fat cottage cheese vs. feta cheese. Food processing is associated with a considerable increase in salt content [11]. Differences in the content of salt can also reflect the local community’s eating customs and taste preferences and partly explain the wide range of the estimated global consumption of salt. Preferred consumption of products containing large amounts of salt can explain why men tend to consume (by as much as one-third) more salt than women [77]. Adolescents also eat more meat and packaged sweets and snacks than recommended, suggesting that these are the sources from which this population derives more salt than other consumer groups [3].

Traditional dishes, cultural aspects and eating habits acquired at home also contribute to increasing sodium intake with the diet. An example is the diet of Malaysians, where the main sources of sodium in the diet were soy sauce, fried rice, omelettes, nasi lemak (rice boiled in coconut milk) and roti canai [48]. Fukutome [81] reported that 15% of total salt consumption volume in Thailand is accounted for by soy sauce and sauces containing soybean paste. Another example is the traditional Mexican diet, where sodium is mostly supplied with tacos [18].

Table 2 presents the highest contribution of various food groups (bread and bakery products, cereals and grain products, dairy, meats, sauces and dressings, seafood, snacks, and vegetables, fruits, nuts and legumes) to daily salt intake in the populations around the world.

## 6. Cereal Products as an Important Source of Salt in the Human Diet

The significance of grains and cereal products in human nutrition is demonstrated by the fact that global food security is mostly determined by the production of cereals, which currently exceeds 2600 million tonnes [83]. Baked goods constitute an important source of sodium since nearly everyone in the world eats baked goods in large amounts [19,20]. Most of the population consumes too much salt, so limiting the intake of sodium from baked goods can contribute to improving their health status. According to Kovac and Knific [84], salt supplied by baked goods can be successfully reduced by 75% without a difference in taste. The WHO intends to collect evidence of the economic aspect of salt reduction and provide advice on eliminating technological barriers and addressing food safety concerns associated with production of high-quality, low-salt baked goods [21].

Laskowski et al. [85] reported that cereal products covered 50% of sodium requirements in Poland. Similarly, our study revealed that cereal products cover 48.8% of an adult’s requirement for Na, of which as much as 48.2% is covered by bread, 0.48% by rice, 0.04% by grains and 0.04% by pasta [23]. Data is also not favourable for the British, for whom cereal products account for 33% of total sodium intake, including 17% from baked goods [86]. For example, if an adult on average eats 250 g of bread a day (equivalent to 8 pieces of bread) containing 1.2% of salt, the consumption of salt from this bread is 3.0 g, which corresponds to 50% of the recommended daily intake of salt. If the bread contains 1.6% of salt, salt consumption is 4.0 g with 250 g of bread, which constitutes 80% of the recommended daily intake of salt [87]. One piece of whole wheat or whole grain bread supplies ca. 0.5 g of salt [11].

According to Syrad et al. [88], the intake of salt from cereal products (bread, breakfast cereals, biscuits, cakes, etc.) accounts for 38% of the daily intake of salt, while the second most significant source of salt in the diet, accounting for 21% of total intake, is meat and meat products. The main sources of salt in the diets of Latvians aged 18–35 years are cereal products, meat and meat products [72]. In contrast, in Norway, meat products are the main source [73]. Americans take most sodium from sandwiches (21%) and cereal products (8%) [89]. In turn, the main source of sodium in Germany is processed food, particularly baked goods, meat, cold meats and cheeses [4], which is similar to Poland (baked goods, cold meats, and cheeses) [39]. For Swiss children, the main sources of sodium were pasta, potatoes and rice (23% of total intake), pastries (16%) and bread (16%) [58]. Among ten products deemed to be the sources of sodium in the diets of Brazilians, four were cereal products, including white bread, rice, pasta and crackers (second, third, fifth and ninth on the list, respectively) [90]. Very similar relationships were observed for the Mexican population, where processed meat was the main component responsible for daily sodium intake, followed by taco, pizza and sweet and salty baked goods [18]. By contrast, in the diets of Costa Ricans, the main sources of sodium were table salt (60%) and processed food and food seasoning (with an addition of sodium) (27.4%) [91]. In the countries of East and Central Asia (Singapore, Malaysia, Philippines, Indonesia, Thailand and Vietnam), sodium in the diet mainly derives from salt and sauces added to the food while cooking, seasoning added on the plate, processed food and snacks (fish balls, fish pancakes, bread, and pasta), as well as beverages [92]. According to available data, bread and dairy products are the main sources of salt in the Eastern Mediterranean region [21]. For instance, in Lebanon, the major dietary contributors to sodium intake among adults were found to include bread (25%), processed meat (12%) and dairy products (10%), such as cheese and labneh (strained yogurt) [93]. Similarly, in Morocco, the main contributors included cereals and cereal-based products, followed by spices and condiments, and milk and milk products [94].

Breakfast cereals, particularly processed ones, can supply considerable amounts of Na, and these products are eagerly consumed mainly by children around the world [6]. Cornflakes, which are particularly popular among Polish children, contain more than 500 mg Na per 100 g of the product (5 g Na per 1 kg). The content of this element in sweetened cereals is also high, with more than 200 mg Na per 100 g (2 g Na per 1 kg) [6]. According to the above-mentioned study, one serving of cornflakes prepared with milk (one serving size = 30 g cereal + 125 mL milk) will cover 14% of schoolchildren’s requirement, supplying ca. 0.2 g of Na, whereas one serving of other cereal will cover 5–17%, depending on its type. According to Daugirdas [95], most cereals available on the market contain ca. 200 mg or even 300 mg of Na per serving, and milk also contains ca. 65 mg of Na. The analysis of the content of Na in two popular US brands of sweetened cereals indicated that, on average, they contained 4.2 g of Na per 1 kg [96], while the average content of Na in 276 analysed cereal brands in Slovenia amounted to ca. 2.3 g Na per 1 kg (228 mg Na per 100 g) [97]. Small natural amounts of Na occur only in natural cereals and bran (wheat, rye, oats, rice), with ca. 10–40 mg Na per 1 kg of dry product [6,95]. It is estimated that the content of sodium in unprocessed raw materials is low and accounts only for 10% of the daily sodium intake [75].

Table 3 presents a ranking of salt content in cereal products, including its share of daily intake in various countries. The data was compiled for seven countries representing four different continents. Rankings of cereal products differ since their consumption depends on people’s eating habits in a specific country and cultural customs determining consumption. However, four out of seven countries put baked goods (including bread and rolls) first in the ranking, and their share of daily salt intake was the highest in Latvia (67%) and the lowest in Switzerland (23.2%). In the USA and Switzerland, most salts were taken daily with pasta (salted and boiled), and in Mexico, with tacos (corn or wheat flour tortilla with beef, onions, garlic, spices and tomatoes). The lowest ranking items were cookies and sweet buns, cereal bars and breakfast cereals and crackers.

Despite considerable differences in the eating habits of adults living in various European countries (reflecting regional differences reported in 2014 by the European Commission) [73], the content of salt in the diets of Europeans can be obviously reduced by implementing changes in basic food categories, such as milk/dairies, meat/meat products, grains/cereal products (including bread). Since cereal products are most often consumed and in the largest amounts around the world, decreasing the amount of salt in bread can largely contribute to reducing salt consumption throughout the population. The WHO [98] established global sodium benchmarks for 18 food categories, including bread and bread products. These benchmarks were developed to call for accelerated action from Member States to scale up their efforts to reduce their populations’ sodium intake. The benchmarks for bread and bread products are as follows: 475 mg/100 g for scones and soda bread, 330 mg/100 g for yeast-leavened breads with all types of cereal flours, 320 mg/100 g for flat breads (tortillas, wraps, naan, pita) and 310 mg/100 g for sweet rolls with raisins/nuts. Some European countries, as well as certain African and Asian ones [21], have successfully implemented measures to reduce salt content in baked goods [99,100]. This data corroborates the significance of the intersectoral approach to changes in food composition and social education on minimising the amount of salt added to homemade meals [101]. The European Commission [73] recommends a 1.0–1.2% salt content. Recently, most countries reduced the amount of salt added to baked goods, yet this limit is sometimes exceeded. Details are presented in Table 4.

## 7. Initiatives to Reduce the Content of Sodium in Food Products and Meals

An excessive intake of NaCl with the diet (>5–6 g/day for adults and >3–6 g/day for children) is observed in most countries of Europe and the world [11]. Results of many years’ analyses of the content of Na in food products in various countries testify to the effectiveness of the gradual reduction-oriented campaigns [4].

The WHO promotes reduction of salt as the best strategy aiming to restrain chronic diseases and Member States approved the target of 30% reduction in average intake of salt by the population by 2025 [105]. The Eastern Mediterranean Region has the highest number of countries in the world (>20%) implementing mandatory salt limits in foods [17]. To date, 5% of Member States (*n* = 9) have implemented at least two mandatory sodium reduction policies and other measures, 22% of Member States (*n* = 43) have implemented at least one mandatory policy or measure, 33% of the remaining Member States (*n* = 64) have implemented at least one voluntary policy and other measures to reduce sodium intake, while 29% (*n* = 56) have made a policy commitment towards sodium reduction [17]. Education at the population level can be more effective in reducing the intake of sodium with the diet than individual interventions [106].

National salt reduction initiatives were multifaceted in approach, characterized by a combination of two or more implementation strategies. Interventions in settings (including interventions targeted at schools and hospitals) were the most common approach. This was followed by food reformulation through engagement with the food industry (established salt targets), consumer education interventions (led by the government, non-government organizations and the food industry), front-of-pack labelling schemes (through warning labels, traffic lights, health messages and percentage daily intake or guideline daily amount) and salt taxation [17].

### 7.1. Changes in Food Products Labelling

Front-of-pack (FOP) nutrition labelling is an efficient strategy allowing consumers to make conscious and healthier food choices [107]. Countries of the world feature different labelling practices with reference to functional and visual aspects, such as type of expression (discretionary or mandatory) and presence of any guidelines for consumers facilitating label interpretation [108]. In some countries, front-of-pack labelling is mandatory (e.g., in Chile, Sri Lanka, Argentina, Indonesia and Finland), but in most of them it is discretionary (e.g., in France, Australia, New Zealand, China and Malaysia) [17]. Labels facilitating nutritional value interpretation show prints, symbols or warnings referring to general content of nutrients in the product. These include, for example, Chilean style warning labels taking note of the content of salt, saturated fat, sugar and the product’s energy value; the Multiple Traffic Light is used in some countries, including the United Kingdom, indicating red (high), amber (medium) or green (low) levels of nutrients (energy value, content of sugar, fat and saturated fatty acids and salt) per serving; and the Health Star Rating used in Australia and New Zealand where the rating from a half to five stars shows if the product is healthy [109,110]. Some labelling schemes (e.g., Guideline Daily Amount in the United Kingdom) convey nutritional content as numbers rather than graphics, symbols or colours, allowing consumers to create their own judgements on healthfulness [111].

On 13 December 2014, Poland put a Regulation of the European Parliament and the Council (EU) No 1169/2011 into effect regarding the provision of food information to consumers. The above-mentioned regulation imposed an obligation—effective as of 13 December 2016—to provide the following information on food packaging: energy value, fat content, saturated fatty acids, carbohydrates, sugars, protein and salt [10,112].

Nutri-Score is a food labelling system implemented in Poland in 2021. Similar Nutri-Scores have been in use for some time in other European countries, including France, Norway, Sweden, the Netherlands, Denmark and Iceland. The European Commission is currently working on making such labelling mandatory. Nutri-Score is a nutrition label using five colours to facilitate assessing whether the product is healthy or not without analysing the lengthy list of ingredients. The scale ranges from the dark green code marked with the letter A to the red code marked with the letter E. Green corresponds to the healthiest products that should be consumed as often as possible. By contrast, E denotes food processed to the highest extent, the consumption of which should be reduced. High-energy products, sugar, saturated fat and salt are negatively evaluated components. In turn, protein, dietary fibre, fruits and vegetables are beneficial ingredients. In our opinion, Nutri-Score should also take note of trans-fatty acids, omega-3 acids, product processing degree and natural additives, such as bioactives and vitamins. The final score is calculated per 100 g or 100 mL of the product, employing a special algorithm that considers the above-mentioned factors and assigns relevant scores. It makes sense but, in some cases, can lead to preferring products consumed in large servings and deprecating those with smaller servings. For example, yoghurt with slightly inferior ingredients and a lower Nutri-Score can be sold in a smaller packaging than its superior counterpart. Therefore, considering typical product servings is a better solution. French scientists developed this system from Santé Publique France to allow consumers to make conscious buying choices [10]. Julia et al. [113] reported that 52% of respondents found Nutri-Score easy to understand compared to other nutrition labelling systems. In addition, 40% claimed that the label helps them choose the right products to buy. The WHO Europe found Nutri-Score to be an important element of the strategy for combating diet-related diseases, which helps consumers make conscious buying choices based on the nutritional value of products. Scientists hope to encourage more people to think about what they put on their plates and incorporate healthier habits in their lives [10]. However, we do notice certain shortcomings of Nutri-Score. For instance, due to energy density and added salt, fatty smoked fish (salmon) is rated lower than certain sweets; dark chocolate (though objectively healthier than milk chocolate) has the same value class (D or E depending on its kind); oatmeal and chocolate cereal are both class A; the system does not distinguish between whole-grain pasta and regular (wheat) pasta or between kinds of rice (brown, white) and groats (coarse, hulled); Nutri-Score reduces the value of high-fat products in two ways (on the one hand on account of their calorific value, and on the other hand on account of fat content, e.g., avocado oil is class D).

In 2022, researchers from the Medical University of Warsaw published a report of their survey [114]. The survey was sent to nutrition and dietetics experts. Only 7% of them agreed that Nutri-Score should be adopted in Poland in its current form. However, the majority of experts (59%) responded that Nutri-Score should be adopted with some modifications. An update of Nutri-Score algorithm is planned to address most of the above-mentioned weaknesses and allow consumers to make real comparisons of foodstuffs [115].

In 2016, under Canada’s Healthy Eating Strategy, Health Canada updated its food labelling regulations. The regulations require that the reference sodium intake coverage be added to nutrition tables on food packaging, assuming 1500 mg/day as the reference value for children’s food and 2300 mg/day for food meant for the adult population [116]. In turn, Canadian regulations establish 60 mg/100 kcal as the maximum amount of sodium in infant formulas [117]. The Healthy Eating Strategy was a supplement and improvement to Canada’s Food Guide that recommends reducing the consumption of processed foods or ready meals with high sodium content. Limitations on food and drink marketing targeted at children and adolescents are still a promising political initiative. In 2022, food nutrition facts labels were enhanced based on consumer opinions. Among other modifications, a note on the percentage content of nutrients (including sodium) in the product was added to help consumers understand how much of the specific nutrient the food contains, indicating that 5% or less is ‘little’ but 15% or more is ‘a lot’ [117].

Since 2009, the Codex Committee on Food Labelling in the United Kingdom has approved including sodium or salt in the list of nutrients to be disclosed on food labels. An international work group was also appointed to discuss which of the two terms—sodium or salt—should be preferably used on the label [118]. At present, two types of nutrition labels exist in the United Kingdom. Firstly, there are voluntary traffic light labels that indicate if the content of saturated fat, sugar and salt is low or high. Red light (product high in the nutrient) recommends careful consumption, yellow light (medium content) raises awareness of an increased risk of health problems, and green light (low content) means that the nutrients are safe to health if standard servings are consumed. Another variant of packaging front labels used in the United Kingdom is GDA (Guideline Daily Amount), which provide information on the total count of calories, fat, fatty acids, sugar and salt in the product and what percentage of the recommended daily intake for adults is satisfied by one serving of the product [79]. A comparison of foodstuffs in the United Kingdom in 2006 and 2011 showed a general average reduction of sodium content by 7%. The sodium content was significantly reduced in ready-to-eat meals, dairy products, sauces and spreads. Decreases in these product groups outweighed increases in the salt content of non-alcoholic beverages and processed vegetables [119]. Thus, voluntary reduction of salt encouraged by the Food Standard Agency contributed to decreasing the overall salt level in food available in the United Kingdom.

On the one hand, the use of FOP labelling on products engages consumers, but, on the other hand, may potentially stimulate food processors to reformulate products to meet nutrition criteria so they can avoid carrying negative FOP labels [120]. In 2016, a black stop sign was introduced in Chile as a comprehensive warning programme covering products with exceeded limits for sodium, saturated fat, total sugars and total energy [121]. A comparison of food labels before and after the first year in which the programme was adopted shows that the percentage of products which should have sodium content warning labels decreased, which implies that food processors reformulated products to avoid a black warning label [122].

Certain limitations on changing food labels do exist. In the study conducted by Haghighian Roudsari et al. [123], insufficient knowledge on label interpretation, small size of back-of-pack traffic light labels and absence of substitutes for red light foods were the principal problems faced by consumers. It is also more difficult to interpret labels if the values are given per serving size instead of 100 g. People are not aware of the definition of serving size, and therefore, using this reference would make the label more complicated. Using a per 100 g reference allows comparisons both within and between food categories. Furthermore, evidence from Australia and New Zealand indicates that the slow uptake by only a small proportion of companies illustrates the limits of commercial goodwill in applying front-of-pack labelling systems voluntarily [17].

### 7.2. Health Campaigns

Bolder actions are still needed to see measurable public health impact. The main WHO goal is to save lives the millions of people. However, the global burden of unhealthy diets constitutes a major public health and development challenge worldwide. In order to increase the benefits for public health, various activities are undertaken, including health campaigns. While such initiatives generate costs, they reduce the availability of unhealthy foods and may result in lower healthcare system costs for chronic diet-related diseases [17]. High-income countries have started to enact public health policies, such as mass media campaigns or limiting salt in processed foods, to reduce the prevalence of hypertension in their populations [124,125]. To date, 96 of 194 Member States associated with the WHO have initiated mass media campaigns focusing on salt intake reduction [17].

The Polish National Food and Nutrition Institute in Warsaw symbolically crosses out the saltshaker on the Pyramid of Healthy Nutrition and Physical Activity and proposes several ways to reduce the intake of NaCl in the diet, such as: limiting the amount of salt added during meal preparation or, if necessary, adding salt at the end of cooking, using fresh or dried herbs instead of salt and choosing products with lower salt content. Special attention is drawn to the use of KCl salt [47].

The British programme Eating Well, Choosing Better, run by the Food Standards Agency, encourages manufacturers to reduce the level of sugar, saturated fat and salt in the food they produce and sell. Salt reduction targets from 2012 were extended in 2017 and finally applied to 76 categories of food. So far, the salt level in many food products has been reduced by as much as 40–50%, and more than 11 million kilograms of salt have been removed from the food. However, the average salt intake in the United Kingdom remains high, amounting to 8.1–8.8 g a day [126].

In 2012, South Korea implemented its National Plan to Reduce Sodium Intake to reduce population Na consumption by 20%, to 3900 mg/day, by 2020. The plan included five key components: a consumer awareness campaign designed to change food consumption behaviours, increased availability of low-sodium foods at schools and worksites, increased availability of low-sodium meals in restaurants, voluntary reformulation of processed foods to lower Na content and development of low-sodium recipes for food prepared at home. It is now known that multicomponent interventions have great potential to reduce population Na intake. The added advantage is that reductions in Na intake were accompanied by reductions in population blood pressure and hypertension prevalence [127]. According to Kweon et al. [128], sodium intake declined significantly over 20 years (4585.6 mg in 1998 and 3255.0 mg in 2018), and notably from 2010. Sodium intake was reduced due to the implementation of the sodium intake reduction policy. The National Plan to Reduce Sodium Intake was a campaign to increase social awareness and optionally reformulate processed foods, including fried pasta, confectionery pastes and products, in order to reduce their sodium content [127].

Alawwa et al. [129] reported that, due to the high intake of sodium by Jordanians, several strategies developed by the WHO to reduce salt intake should be adopted, such as identifying the main sources of sodium in the diet, changing the composition of some food products available on the market, preparing information materials on health to promote the consumers’ awareness of salt and informing them of how to read and interpret food labels.

Considering the alarming data from across the world, the WHO [130] presented its recommendations under the Global Action Plan for the Prevention and Control of Non-Communicable Diseases (2013–2020), and according to these recommendations, sodium in the diet should be reduced to less than 2000 mg or 5 g of salt per person and the level of salt/sodium added to food (prepared or processed) should be reduced by 2020, which will decrease the risk of developing hypertension, i.e., the main CVD and renal failure risk factor. However, despite continuing efforts, the latest data implies that salt consumption is twice the limit recommended by the WHO [17]. More coordinated efforts are needed to develop efficient national salt reduction programmes, maintain high-quality monitoring, and implement policies and interventions known to reduce salt intake at the population level effectively. Even a slight decrease in salt intake in a population has a material impact on human health, and many small measures taken together can make a big difference [131].

The feasibility and effectiveness of health campaigns is not normally verified before and after their implementation, and hypertension treatment and control rates are still unsatisfactory [125].

### 7.3. Salt Use Limitations in Food Manufacturing

Urgent action is required to modify the production and consumption of foods and beverages, including the manufactured food industry. Mandatory maximum limits for sodium in processed foods promote industry-wide reformulation and create a marketplace that restricts the least healthy food options regardless of where people shop or how much they understand and have access to information on labels. The aim is to provide industry with incentives to reformulate and produce healthier products [17].

In 2022, the WHO developed benchmarks for sodium content in 18 food categories and called on food operators to implement them globally [98]. Some of the large food manufacturers have committed to taking steps to achieve those benchmarks, but again, bolder action and engagement from more actors is needed to see measurable public health impact. In total, 34% (*n* = 65) have implemented policies to reformulate manufactured food to contain less sodium through mandatory (11%), mandatory and voluntary (3%), or voluntary (20%) approaches. Reformulation is most commonly implemented in the WHO Eastern Mediterranean and European regions, and is more common in the higher income group. Bread and bread products are the most targeted food category for sodium reduction, followed by processed meat, poultry, game, fish, ready-made and convenience foods, and savoury snacks [17].

Bread is the main candidate for changing the product’s composition (36%), followed by processed meat (28%) and ready meals (23%) [105]. As recommended by the European Commission, the amount of salt used in the production of baked goods should be reduced to 1.0–1.2% per 100 g of flour, and the Member States are required to inform the public about the content of salt in food and the reduction of its levels [73]. Mello et al. [90] report that the content of sodium in baked goods has been recently lowered in Brazil. Analysis of the content of sodium in baked goods sold in Poland in 2017 showed that bread provides 7.59 g of Na per 1 kg [23]. More current data is not available, but the Polish market shows an upward trend in contrast to certain other countries. Research needs to continue to assess the current content of salt in baked goods sold in Poland.

At an individual level, nutritional guidelines for the Brazilian population assume a moderate sodium intake. Moreover, at the population level, the Brazilian Association of Food Industries referred to those guidelines in creating a voluntary agreement that involves more than 70% of the market of sodium-based processed food for specified categories of food: baked goods, pastries, cookies and biscuits, pasta, spreads, breakfast cereals and mayonnaise. Sodium reduction targets in these products improved as sodium levels were gradually reduced in 2011–2017 [132].

Food reformulation is rated as the intervention with the highest priority in terms of effectiveness, equity, sustainability and acceptance by policy-makers [107]. Several previous systematic reviews of food reformulation interventions showed reductions in salt consumption across an entire population [106]. However, structural features, such as voluntary or mandatory nature, can determine the success of food reformulation interventions [133]. Voluntary approaches to food reformulation depend on strong government leadership, extensive advocacy activities, cross-industry collaboration and robust monitoring of salt content in selected products, and most importantly, publishing the results in order to hold the food industry accountable [134].

### 7.4. Taxes on High Salt Content Products

Only one Member State, Hungary, has had an excise tax since 2011 targeting foods high in sodium through underlying nutrient profile modelling. In Portugal, a proposed tax on salt-rich foods was considered in 2018 but was ultimately not approved by the Parliament. Hungary imposed taxes on salty snacks with salt content > 1 g/100 g and on food seasoning with salt content > 5 g/100 g. This tax amounts to 0.8 €/kg of salt. The tax contributed to reducing salt content in many food products by, in some cases, as much as 85%. The effects of salt taxation included a reduction in the sale and consumption of salty snacks by 26% [135]. The main reason behind the change in consumer behaviour was the increase in prices and raised awareness of the negative impact of salt on health [73].

The possibility of imposing taxes on food of little value, such as high-sodium food, may be promising as it will no longer be readily available or affordable [136]. In turn, this can offset the prices of salt substitutes as the demand for them will increase. Although, as a rule, salt substitutes are about 50% more expensive than ordinary salt, they are still very cheap. Salt substitutes are thus available to most patients and seem a good supplement to blood pressure control medication [134]. With time, the use of substitutes can modify the population’s taste preferences and consumption norms. Nevertheless, the content of salt in food reduced by the food industry will most likely reduce the intake of sodium by the population to a higher extent than high-sodium-food taxes [136].

### 7.5. Use of Salt Substitutes

If the amount of salt in food changes, steps should be taken to maintain product acceptability. Usually, this is performed by altering the chemical profile using salt substitutes or boosters, maximum stimulation of salt receptors by ensuring the quick release of salt from food surface, and decreasing consumer preferences regarding salt. The human salt taste receptors can adapt and become more sensitive to low salt concentrations within only 4–6 weeks; therefore, a small gradual decrease in the sodium content of processed food cannot be detected. This means that foods with lower salt content will taste as salty as highly salted foods before the adjustment. Furthermore, evidence indicates that, once salt intake has been reduced, consumers prefer foods with less salt [107]. For instance, in the United Kingdom, salt content in major brands sold in supermarkets reduced by 20–30% over three years did not affect their sales or consumer preferences [100].

Several countries (including the United Kingdom) have considered replacing table salt with salt substitutes containing less sodium, that is, potassium-based salt replacers (KCl) or other similar equivalents, such as magnesium (MgCl_2_, MgSO_2_), as a potential blood-pressure-lowering strategy [137]. Diets higher in sodium and lower in potassium have been considered a leading factor for the development of hypertension. The majority of junk foods and processed and packaged foods have higher sodium contents, and higher sodium and lower potassium dietary intake has become a serious global health challenge. The adverse ratio of both electrolytes is strongly linked to blood pressure and the dietary Na:K ratio is an independent risk factor for metabolic syndrome. Furthermore, modify ratios, including lower Na intakes and higher K intakes (e.g., replacing NaCl with KCl), was suggested as a strategy to prevent metabolic disorders, including hypertension [138]. The meta-analysis of randomised controlled studies showed that salt substitutes low in sodium compared with common salt reduced mean systolic blood pressure by 7.81 mmHg and diastolic blood pressure by 3.96 mmHg. The effects in hypertensive, normotensive and mixed populations were similar [139]. Using salt substitutes (65% NaCl, 25% KCl and 10% MgSO_4_) was found to be a feasible and effective dietary approach to reducing salt intake in the population of China, which can significantly help restrain cerebral stroke prevalence among the Chinese people living in rural areas by up to one million cases every year [134]. In the above-mentioned study, systolic pressure in hypertensives was reduced within 12 months by 4.0–5.4 mmHg; however, diastolic pressure did not change.

Gusmão et al. [22] examined the quality and flavour of baked goods in which KCl partially substituted NaCl with satisfactory results with regard to both rheological aspects and flavour evaluated by consumers. A bread dough recipe containing 1% NaCl and 0.5% KCl reduces the content of Na in the product by 56% without significantly compromising the quality of bread. In turn, Bassett et al. [140] partially substituted (50, 70 and 80%) NaCl by a mixture of CaCl_2_ and CaCO_3_ (1:1 ratio). Those authors found that bread containing 50% of NaCl was not different in organoleptic properties from bread that contained only table salt. However, there is a risk that a reduction in the content of NaCl for KCl (>30%) can lead to an increase in the relative content of Mg and K, which can lead to a metallic taste and bitterness of bread that are not acceptable to consumers [141].

In studies exploring taste acceptability of six different potassium-enriched salt substitutes, more than 80% of individuals did not differentiate between regular salt and potassium-enriched salt substitutes containing less than 30% KCl [142]. The risks of potassium-enriched salt substitutes include a possible increased risk of hyperkalaemia and its principal adverse consequences: arrhythmias and sudden cardiac death, especially in people with conditions that impair potassium excretion, such as chronic kidney disease [143]. However, there is not enough evidence regarding the effects of potassium-enriched salt on serum potassium levels and the occurrence of hyperkalaemia in people with chronic kidney disease and others at risk for hyperkalaemia [144]. Five studies from meta-analysis conducted by Tsai et al. [145] provided the data of follow-up serum potassium level. Using salt substitute increased the serum potassium by 1 mmol/L, but it did not achieve significant elevation in serum potassium level. This is confirmed by other studies. In a study by Yuan et al. [146] noted clear benefits of a salt substitute (KCl) compared to NaCl in lowering blood pressure as well as protecting against cardiovascular events among elderly people living in nursing homes in China. These benefits were accompanied by an increase in blood chemistry hyperkalaemia, but there was no evidence of adverse clinical outcomes. In turn, in the study by Yu et al. [147], conducted among Indian hypertensive patients and where NaCl was also replaced with KCl, no adverse clinical events, including hyperkalaemia, were observed.

Kremer et al. [148] tested the palatability of bread in which soy sauce was the only source of Na, apart from Na naturally occurring in grains, with no additions of Na in forms other than salt, for example, monosodium glutamate. As a result, the salt level in bread was reduced by 38.9%. According to consumers, such bread was no less attractive than the standard one. The quality and acceptability to consumers were also maintained for other food products (such as cheese or processed meat products) if the content of salt was reduced gradually [149,150,151,152].

Approaches to using potassium-enriched salt substitutes as the main alternative may vary from country to country, depending on food preferences and the source of sodium in the population’s diet. In countries such as the United States, where 70% of the total sodium intake derives from commercially processed food and meals consumed in restaurants, food reformulation can be a particularly effective sodium reduction strategy [153]. By contrast, replacing common salt added to meals cooked at home can be more effective in countries such as China, where even three-quarters of sodium intake derives from salt added to cooked meals [154].

Another way to increase the perceived saltiness of food is salt flavour enhancers. For instance, organic acids such as lactic, citric and acetic acids and some amino acids (arginine, aspartic acid, lysine and glutamate) enhance salty taste [155,156]. Moreover, monosodium glutamate and yeast products can be used for enhancing salty taste, often in combination with KCl [156].

### 7.6. Avoiding Adding Salt for Cooking

Although pasta contains less than 200 mg Na per 100 g and thus is not considered a high-sodium food itself, it is often served with sauces, which can dramatically increase its Na contents upon its consumption [65]. It is noteworthy that significant and varying Na content results from salt added during preparation. In our study, our team noticed that salt added to water while cooking pasta increased the content of Na nearly five times compared to pasta cooked in unsalted water. In turn, using water without salt reduced the content of Na in the cooked pasta by 48% compared to its content in pasta cooked in water with 3.17 g of salt and by 99% compared to its content in pasta cooked in water with 6.34 g of salt [157]. According to Bianchi et al. [158], due to increasing the content of Na in water by adding salt, Na from the solution penetrates the product, thereby reducing the content of other minerals, while cooking in unsalted water causes Na to penetrate from the product into the water. The degree of penetration of minerals from pasta into the water and vice versa during cooking depends on the content of Na in water. Reducing (or eliminating) the amount of salt added when cooking pasta is a quantitative and straightforward way to reduce dietary Na. It could also be communicated that rinsing after cooking could reduce by one-third the Na content of pasta cooked in salted water, but other minerals, such as Zn, Mn, K, Cu, Fe, Ca and Mg, are flushed at the same time [157,158]. Only additions to pasta, such as sauces, should be seasoned, whilst one should reduce the amount of salt when seasoning, and instead of salt, use herbs and natural spices for flavour [157]. Because Na in the form of table salt is very soluble in water and easily diffused, the recommendation not to add salt to the water when cooking may also apply to other foodstuffs, such as eggs, rice, groats, vegetables, and legumes [54,157].

In addition, effective ways of reducing the amount of salt in the diet are reducing product serving size, using low-sodium products, and controlling the use of intermediate goods in households. Figure 3 presents selected strategies to reduce Na intake from foods.

## 8. Perspectives and Conclusions

In 2016, the FDA acted on the IOM’s recommendation by proposing federal recommendations for phased reductions in Na content for about 150 food categories. Some food companies, such as Danone North America, Mars, Nestlé, PepsiCo and Unilever, agreed that reducing Na levels can be important for public health and that food companies should do more to reduce Na in their products [159,160]. International authorities on human health support the idea that reducing salt content in food is a viable strategy for improving public health status. Food industry sectors are engaged in limiting salt intake despite the fact that changing product composition can entail additional costs of product development. Therefore, businesses from the food sector should be encouraged to put healthy diet components on the market. In the first place, this can be performed through selective taxation and subsidies, marketing control and regulations on food quality, and—from a longer perspective—consumer wishes to combine healthier food with a continuous pursuit of comfort [161]. Furthermore, through their Live-Well programme, the USA attempts to ensure that participating restaurants offer healthy food to children and adolescents, encouraging them to eat more fruits, non-starchy vegetables and wholegrain products, and to limit the consumption of added sugar and sodium [162]. The content of NaCl in the diet should be reduced not only by food producers but also by changing the population’s eating habits and nutrition education [163]. However, collaboration from all parties is necessary to reduce salt consumption efficiently; it is not enough to reduce salt content in food products. Consumers should be encouraged to buy low-salt products and limit salt intake at their own discretion [164]. Figure 4 presents the effects of excessive consumption of salt and the benefits of its reduced intake.

A review of studies revealed numerous national and international salt reduction strategies worldwide. Many countries (in Europe and globally) developed strategies to reduce salt intake and strive to reduce salt consumption using behavioural and structural preventive measures gradually. However, despite their numbers growing, none of these countries has achieved the target of 30% relative reduction of salt intake by 2025 [135]. To date, not every Member State has implemented rapid strategies of government-led and comprehensive mandatory sodium reduction policies and other measures. This may be due to insufficient awareness of the governments about the significant impact of reducing dietary sodium on human health and life. In our opinion, the WHO report from 2023 [17] will encourage countries to fulfil obligations and contribute to saving millions of human lives. The WHO proposes that, if all countries accelerate policy adoption to ensure at least two mandatory interventions and implementation of best practices, it is possible to dramatically reduce salt intake. Therefore, whether to extend the target to 2030 is being considered. The analyses by Hendriksen et al. [44] show that, compared to other European countries, Poland will derive more benefits from reducing salt intake; a 13.5% reduction in cerebral stroke occurrence is expected, while in Finland, this number will be 6.4% only. Institutions in charge of these measures must accelerate their efforts, regularly monitor and evaluate their strategies, and share their findings of efficient measures with other countries to ensure that they will achieve the global target together. The mission to reduce salt, both hidden in food products and added when preparing meals, will allow for the improvement of eating habits in the youngest generation and prevent deaths from diet-related cardiovascular diseases. Furthermore, it will contribute to decreasing the consumption of processed foods (accounting for about 75% of daily intake of salt in many high-income countries).

It is necessary to develop and effectively promote relevant strategies and implement them, e.g., by means of media campaigns, as well as to adjust legislation regarding acceptable levels of Na in the most popular products (i.e., most frequently consumed ones). In turn, the fact that information about the content of Na is often missing from food labels is an inconvenience to consumers [23]. The findings of Korošec and Pravst [97] were similar, as only 1% of labels on more than 5000 analysed foodstuffs in Slovenia displayed such information. A need also exists to educate society at the national level about recommended sodium intake with the diet and, most importantly, about sources of sodium in the diet, to raise the population’s interest and motivate the commitment to reduce salt. Individual consumers should be encouraged to use as little salt as possible in preparing their meals (behavioural prevention) and avoid adding salt on the plate [4]. Instructing society not to add salt in cooking is also essential, along with the controlled use of intermediate products in households.

To reduce salt intake throughout the population, processed food, such as cereal products, meat, cold meats and cheeses, should contain less salt. In view of food technology, the most important and the easiest change to make is reduce the content of salt in baked goods and other cereal products. A reduction in the salt content from 1.2% to 0.6% or even 0.3% had no significant effect on the rheological and physical properties of wheat bread dough. This implies that salt content reduced by half does not significantly reduce the flexibility of bread dough [22]. The biggest opportunity for change in the near future is provided by a moderate reduction of salt content in baked goods, primarily through the partial use of salt substitutes. Further studies on efficient methods of reducing sodium content by changing the ingredients of food products and processing methods are indispensable to facing the challenges of safe food production.

There is the risk of a discrepancy between salt intake calculated based on nutrition interviews or surveys and the actual supply based on 24-h excretion of sodium with urine [105]. However, more reliable studies based on 24-h excretion of sodium with urine are not readily available in daily medical practice. Therefore, it seems that hypertensiologists need to be supported by nutritionists. Considering that the choice of adequate food products and skilful replacement of salt with substitutes is a particularly important issue for reducing salt intake, efforts should be taken to increase the involvement of this professional group in educating patients.

## Figures and Tables

**Figure 1 nutrients-15-02393-f001:**
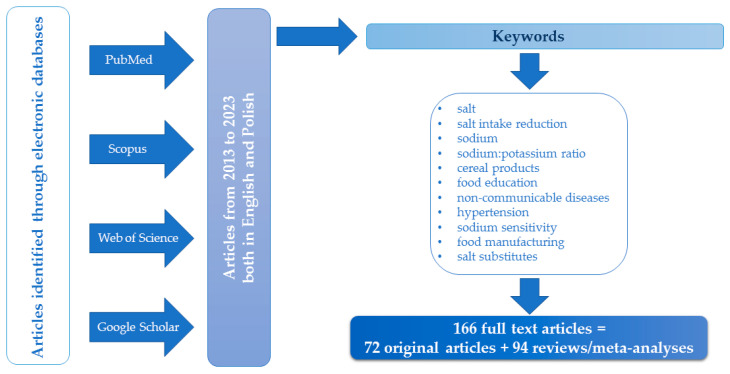
Research strategy employed in the review of available literature.

**Figure 2 nutrients-15-02393-f002:**
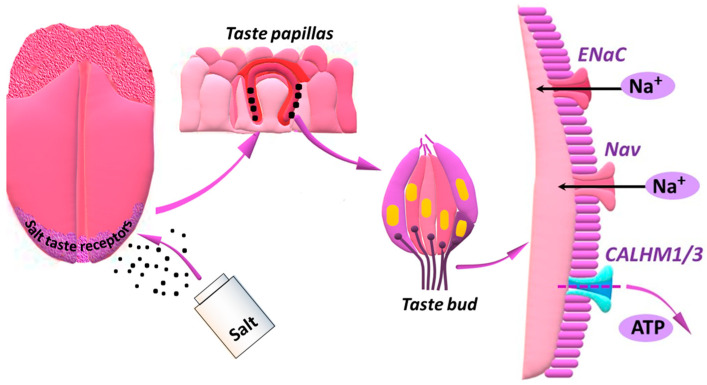
Cells and signal transduction for salty taste. ENaC—epithelial sodium channel; CALHM1/3—calcium homeostasis modulator 1/3; ATP—adenosine-5′-triphosphate; P2 × 2/3—ATP receptors.

**Figure 3 nutrients-15-02393-f003:**
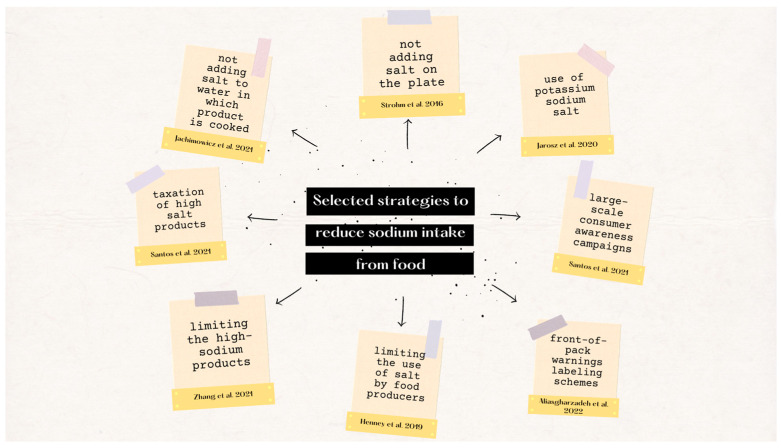
Selected strategies to reduce sodium intake from foods [4,47,65,107,135,157,159].

**Figure 4 nutrients-15-02393-f004:**
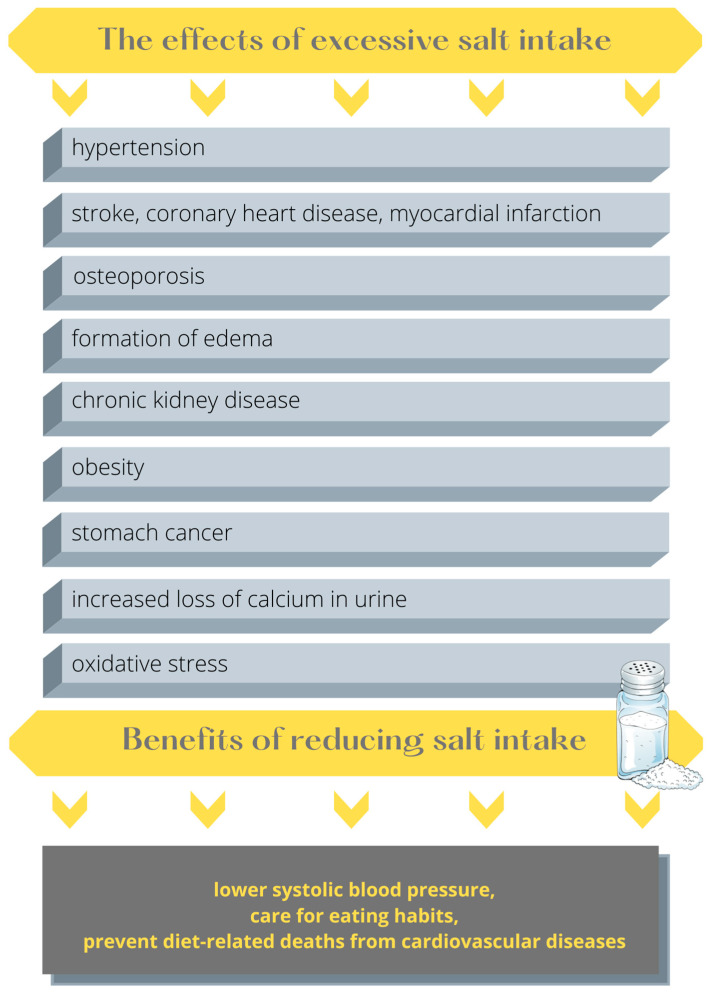
Effects of excessive salt intake and benefits of reduced salt intake.

**Table 1 nutrients-15-02393-t001:** Selected sodium (Na) intakes and salt (NaCl) equivalents in various countries of the world (WHO Member States), based on [17].

Country	Intake of Na [mg]	NaCl Equivalent ^1^ [g]
The highest intake		
China	6954	17.7
Hungary	5646	14.3
Czechia	5112	13.0
Bulgaria	5087	12.9
Croatia	5077	12.9
The lowest intake		
Samoa	2006	5.1
Türkiye	2071	5.3
Democratic Republic of the Congo	2236	5.7
Estonia	2259	5.7
Syrian Arab Republic	2367	6.0

^1^ the content of salt and sodium in products or diets should be calculated based on the assumption that 1 g of sodium corresponds to about 2.5 g of salt [11].

**Table 2 nutrients-15-02393-t002:** Contribution of various food groups to daily salt intake using data available from Bhat et al. [82].

Food Groups	Countries with Highest Contribution of Various Food Groups to Daily Salt Intake	% Contribution
Bread and bakery products	USA, Spain, Poland	34.9–41.1
Cereals and grain products	China, Brazil, Switzerland	23.7–23.1
Dairy products	New Zealand, Argentina, Poland	15.4–14.1
Meats products	USA, Finland, Poland	31.2–30.8
Sauces and dressings	Japan *	44.3
Seafood	Japan, China, Finland	10.2–9.0
Snacks	Argentina, Canada, Mexico	6.9–5.5
Vegetables, fruits, nuts and legumes	Brazil, Poland, Japan	18.8–18.4

* One country (Japan) is given due to its highest contribution compared to other countries.

**Table 3 nutrients-15-02393-t003:** Ranking of salt content in cereal products and its percentage share in the daily intake in various countries.

Country	Ranking of the Content of NaCl in Cereal Products	Percentage of Daily NaCl Intake [%]	References
Latvia	Bread and flour products > breakfast cereals and oatmeal porridge > biscuits, cakes and bread rolls	15–67	[72]
USA	Pasta > rice > taco and burrito > pizza > savoury snacks > sweet snacks > yeast breads and tortillas > breakfast cereals and bars	3–21	[89]
Poland	Bread > snacks (breadsticks and crackers) > breakfast cereals	1.1–23.7	[39]
Switzerland	Pasta and rice > pastries > bread	16.2–23.2	[58]
Brazil	Bread > rice > pasta > crackers	2.59–12.38	[90]
Australia	Bread and bread rolls > mixed cereal dishes > pastries > breakfast cereals and bars > savoury biscuits > muffins	2.0–13.4	[20]
Australia	Bread and rolls > breakfast cereal > mixed dishes with cereals	2.9–10.9	[45]
Mexico	Tacos > pizza > sweet bakery > savoury bread > breakfast cereal > wheat flour tortillas > crackers > cookies and cereal bars	2.9–21.8	[18]

**Table 4 nutrients-15-02393-t004:** The average content of NaCl in bread in various countries.

Country	Average Content of NaCl in Bread [%]	References
Latvia	0.8–1.2	[87]
UK	0.95	[100]
Germany	1.6	[101]
Ireland	0.6–1.68	[102]
Poland	1.64	[39]
Switzerland	1.0	[58]
Netherlands	1.19	[103]
Italy	1.5	[104]

## Data Availability

Not applicable.

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
