# Peer review of "Initiatives to Reduce the Content of Sodium in Food Products and Meals and Improve the Population’s Health"

_nutrients, 2023, doi:10.3390/nu15102393_

Round 1

Reviewer 1 Report

This is a nice review and a good summary of some of the challenges of salt reduction in foods.

I just had a few (minor) issues:

1. The paper tries to cover a wide range of different food categories in the main, but has a specific section focused on cereal products (section 6)?  I'm not sure if the paper would benefit from having a stronger focus just on cereal products (and a different title) or if the intention is to keep this broader to have additional sub-headings and sections for other categories of foods.

2. Do take-away foods/out-of home foods deserve an even stronger mention (in section 5) including specific mention earlier in this section (section 5).

3. The importance of sodium:potassium ratio (e.g. in hypertension) doesn't seem to get a mention?

Well written paper - no issues 

Reviewer 2 Report

Dear Authors, 

Thank you for your paper which has lots of information, however, it is difficult to follow the flow of information/argument for or against the effect of salt reduction on population health, as the sections are often repetitive, overlapping and argument unclear. 

The methodology of the review is required, as well as updating a number of references to the most recent data. 

Additional information is needed on the impact of the interventions on the food manufacturing industry, cost/benefits for public health, why do the authors believe the 30% reduction target has not been met, is the multifaceted approach beneficial or not, what are the recommendations to help countries/populations achieve a 30% reduction?

Please find comments below for your consideration:

Line 44 - There are more recent data/references on CVD death associated with excessive sodium intake - see GBD 2019 data.

Line 48 - WHO spelling is Organization

Line 51 - Please provide a reference for 6 g day (for food making purposes)

Line 69 - Please define the initiatives, and add references.

Line 70 - The cost to food manufacturing is noted, the cost to public health of excess sodium should also be noted. There are a number of studies which outline the cost to society and the economy. 

Line 78 - To be placed in a methods section, please include search terms, inclusion / exclusion criteria, grey literature  - english only?, regions? age groups? etc number of reviewers, and flow chart for the total of 172 publications.

Line 170 - Consider using GBD2019 data, this data has been used by WHO in the NCD data portal as well as the Sodium Intake Reduction report 2023 which provides intakes for all 194 Member States, and by WHO Regions and World Bank income levels. This information should replace lines 170-202 with a practical approach to demonstrate sodium intake across the world and Table 1.  The data provided is not a comprehensive representation. 

Line 232 - check GBD2019 data.

Line 234-245 should this information be placed with lines 2017-225 which focus on infants and children?

Line 226 - consider adding more evidence on the impact of excess sodium intake CVD and death, return on investment on reducing sodium, there is a wealth of detail which could be added. 

Line 262 - there is potential to include data on low sodium salt substitutes in this section. 

Line 280- could a table be added to this section to demonstrate the sources of sodium in the diet around the world. Bhat S, et al have published on this, and at minimum a reference to their work should be included.

Line 432 - Check the WHO Global Report on Sodium Intake Reduction 2023 for updated information re country implementation numbers.

Line 448 - See above report for FOPL data which also includes mandatory and voluntary information

Line 547 - Could you provide more information on the challenges, cost to businesses/organizations, consumer acceptance/recognition etc 

Line 555 -  consider updating ref119 

Line 597 - check reference for most updated data

Line 606 - the WHO benchmarks for sodium reduction should be added to this section

Figure 3 -  please explain "care for eating habits"? also diet-dependent, should that be diet-related? reduce the consumption of processed foods, seems out of place with the disease outcomes above.

Line 789 - could the authors offer a perspective or opinion as to why the target has not yet been met?

Line 815 - why have the WHO Benchmarks for reducing sodium in baked goods not been referenced and discussed?

Line 822 - salt substitutes, if this is a recommendation/finding, please include the negative effects within the impact section. Will salt substitutes make a substantive change to the sodium content of processed foods? The literature provided was from China in which the major source of sodium in the diet is added to food through salt and/or sauces. Note the substantive data from India.

Round 2

Reviewer 2 Report

Dear authors, 

Congratulations on your paper, and addressing the comments provided for your consideration.

Author Response

Response to Round 2 Reviewer 2 Comments

Thank you again for valuable comments which have enriched the paper.

Best regards!